

# Deconvolute individual genomes from metagenome sequences through short read clustering

Kexue Li[1,2,*], Yakang Lu[1,2,*], Li Deng[1,2,3], Lili Wang[1,2], Lizhen Shi[4] and Zhong Wang[3,5,6]

[1] School of Mechanics Engineering and Automation, Shanghai University, Shanghai, China
[2] Shanghai Key Laboratory of Power Station Automation Technology, Shanghai, China
[3] Department of Energy, Joint Genome Institute, Walnut Creek, CA, USA
[4] Department of Computer Science, Florida State University, Tallahassee, FL, USA
[5] Environmental Genomics and Systems Biology Division, Lawrence Berkeley National Laboratory, Berkeley, CA, USA
[6] School of Natural Sciences, University of California at Merced, Merced, CA, USA
* These authors contributed equally to this work.

## ABSTRACT

Metagenome assembly from short next-generation sequencing data is a challenging process due to its large scale and computational complexity. Clustering short reads by species before assembly offers a unique opportunity for parallel downstream assembly of genomes with individualized optimization. However, current read clustering methods suffer either false negative (under-clustering) or false positive (over-clustering) problems. Here we extended our previous read clustering software, SpaRC, by exploiting statistics derived from multiple samples in a dataset to reduce the under-clustering problem. Using synthetic and real-world datasets we demonstrated that this method has the potential to cluster almost all of the short reads from genomes with sufficient sequencing coverage. The improved read clustering in turn leads to improved downstream genome assembly quality.

## INTRODUCTION

Metagenome sequencing holds the key to comprehensively understand the structure, dynamics and interactions of underlying microbial communities and their implication to health and environment (*Chiu & Miller, 2019*; *Tringe & Rubin, 2005*; *Thomas, Gilbert & Meyer, 2012*). As these samples often consist of thousands of different species with highly uneven richness, exceptional sequencing depth is required to study relatively rare species. As a result, except for a few cases (*Brown et al., 2017*), the majority of metagenome sequencing projects relied on cost-effective, short-read sequencing technologies. These projects routinely produce a huge amount of data of 100–1,000 giga-bases (Gb) or more (*Howe et al., 2014*; *Shi et al., 2014*). The largest project so far is the Tara Ocean Metagenomics project, where 7.2 tera-bases (Tb) was generated and the Prokaryote subset alone contains 28.8 billion short reads (*Sunagawa et al., 2015*).

Corresponding authors
Li Deng, dengli@shu.edu.cn
Zhong Wang, zhongwang@lbl.gov

As the majority of members of these microbial communities are not known, assembling the short reads into draft genomes, or metagenome assembly, is a key step in metagenomics. Metagenome assemblers have to deal with both scale (billions of short, 100–250 bp reads) and complexity problems (thousands of different species with a highly uneven abundance distribution). Most assemblers first assemble the short reads into longer contigs, then cluster the contigs into individual draft genomes through the binning process (*Roumpeka et al., 2017*; *Kang, Rubin & Wang, 2016*). The assembly step in these software tools simultaneously tackles the computational and algorithmic challenges by constructing a huge *de Bruijn* graph and subsequently partitions it in parallel (reviewed in *Breitwieser, Lu & Salzberg (2017)* and *Quince et al. (2017)*). These tools, including MEGAHIT (*Li et al., 2015*), metaSpades (*Nurk et al., 2017*) and MetaHipmer (*Georganas et al., 2018*), have achieved considerable success and are widely used. To overcome the limitation of this "assembly-then-cluster" approach that does not allow optimization for individual genome assembly, a "cluster-then-assembly" alternative has recently been proposed. This strategy first clusters the reads based on their genome of origin (*Guo et al., 2015*; *Shi et al., 2018*), and then each cluster can be assembled individually and potentially optimized. Tools adopting this strategy take advantage of the scalability and robustness of Apache TM Hadoop (*Guo et al., 2015*) or Spark (*Shi et al., 2018*) platforms to construct and partition an overlap graph in parallel.

We previously reported that an Apache SparkTM-based read clustering method, SpaRC, that showed a great potential in achieving good scalability and clustering performance (*Shi et al., 2018*). SpaRC can be flexibly deployed to the cloud or HPC computing environments. However, the demonstrated clustering success was limited to long-read sequencing technologies. Even though SpaRC can form pure clusters (low false positives), clustering short-read datasets suffered a false negative problem, or one genome is clustered into many small clusters (under-clustering). This is not desirable as most of the metagenome datasets are based only on short-read sequencing technologies. Clustering short reads to recover single genomes has been previously shown to be possible by a latent strain analysis approach (LSA, *Cleary et al. (2015)*). However, clustering metagenome reads directly based on $k$-mer statistics across multiple samples is very challenging (*Wang et al., 2012*; *Liao et al., 2013*).

In this article, we describe a new method to target the under-clustering problem of SpaRC by exploiting statistics derived from multiple, independent samples in short-read datasets. This method first estimates the abundance of each read cluster using a set of short, representative $k$-mers, and then calculates the similarity among the clusters and uses it to construct a graph of clusters. Finally, it partitions the cluster graph to obtain larger read clusters. We name the new clustering algorithm developed here as "global clustering", as it deals with cross-sample information from the entire dataset. Conversely, the clustering algorithm in SpaRC we had reported previously is now renamed as "local clustering", as it only deals with read overlap information. We implemented the global clustering algorithm on the Apache Spark platform to achieve data scalability and computing robustness. In addition, we adopted minimizers (*Roberts et al., 2004*) as a replacement for $k$-mers to improve computing and memory efficiency. We compared the

clustering performance of the global clustering algorithm to the local clustering using a synthetic mouse gut microbiome dataset from the CAMI2 project (*Sczyrba et al., 2017*). Several clustering parameters were also explored to gauge their influence on global clustering performance. Using a real-world metagenome dataset, we showed that clustering the reads before the assembly can greatly improve the assembly quality of the species with high sequencing coverage.

## MATERIALS AND METHODS

### Clustering strategies

An overview of the two clustering strategies is shown in Fig. 1. The local clustering strategy in the original SpaRC has been described in *Shi et al. (2018)*. In brief, during the local clustering step, we cluster reads by their overlap. The read clusters are further clustered into bigger clusters by the global clustering strategy, which we will describe in detail below.

### Local clustering improvement with minimizers

In SpaRC, the number of shared $k$-mers is used to estimate similarity between reads (*Shi et al., 2018*). As it takes 100–200 times more space after reads are transformed into $k$-mers and edges, SpaRC is neither space nor time-efficient. To improve computing efficiency, we implemented a new function to use minimizers (*Roberts et al., 2004*) instead of $k$-mers to estimate similarity between reads. As many adjacent shared $k$-mers can be represented by a single minimizer without losing sensitivity, in theory, the minimizer-based method should greatly reduce the memory requirement in SpaRC (as fewer $k$-mers and edges will be produced). In practice we did observe a 3.2-fold memory usage reduction, and 3.3-fold speed-up (Fig. S1). It is worth noting that minimizers may not be applicable to uncorrected long reads from PacBio and Nanopore sequencing technologies due to their high error rate.

### Global clustering

Reads from a genome can form many read clusters after the local clustering step, leading to low clustering completeness. The ultimate goal of global clustering is to predict all the read clusters originated from the same genome. It does this based on the assumption that the sequencing coverage of each region of a genome, defined by the read clusters, closely resembles the sequencing coverage of the same genome across different metagenomic samples. In other words, if two clusters, $c1$ and $c2$, belong to the same genome $g$. After $c1$ and $c2$ are assembled into contigs $C1$ and $C2$, the coverage of $C1$ and $C2$ in sample $S$, in theory, should be equal to the coverage of $g$.

### *Estimating the sequencing coverage of an underlying genome based on a cluster of unassembled reads*

In the context of single genome assembly, the sequencing coverage of a genome can be robustly estimated from unassembled reads by $k$-mer analysis (*Chor et al., 2009*; *Lo & Chain, 2014*). Similarly, we can estimate the sequencing coverage of the latent genome represented by a read cluster. As shown in Fig. 1B, clusters from different genomes,
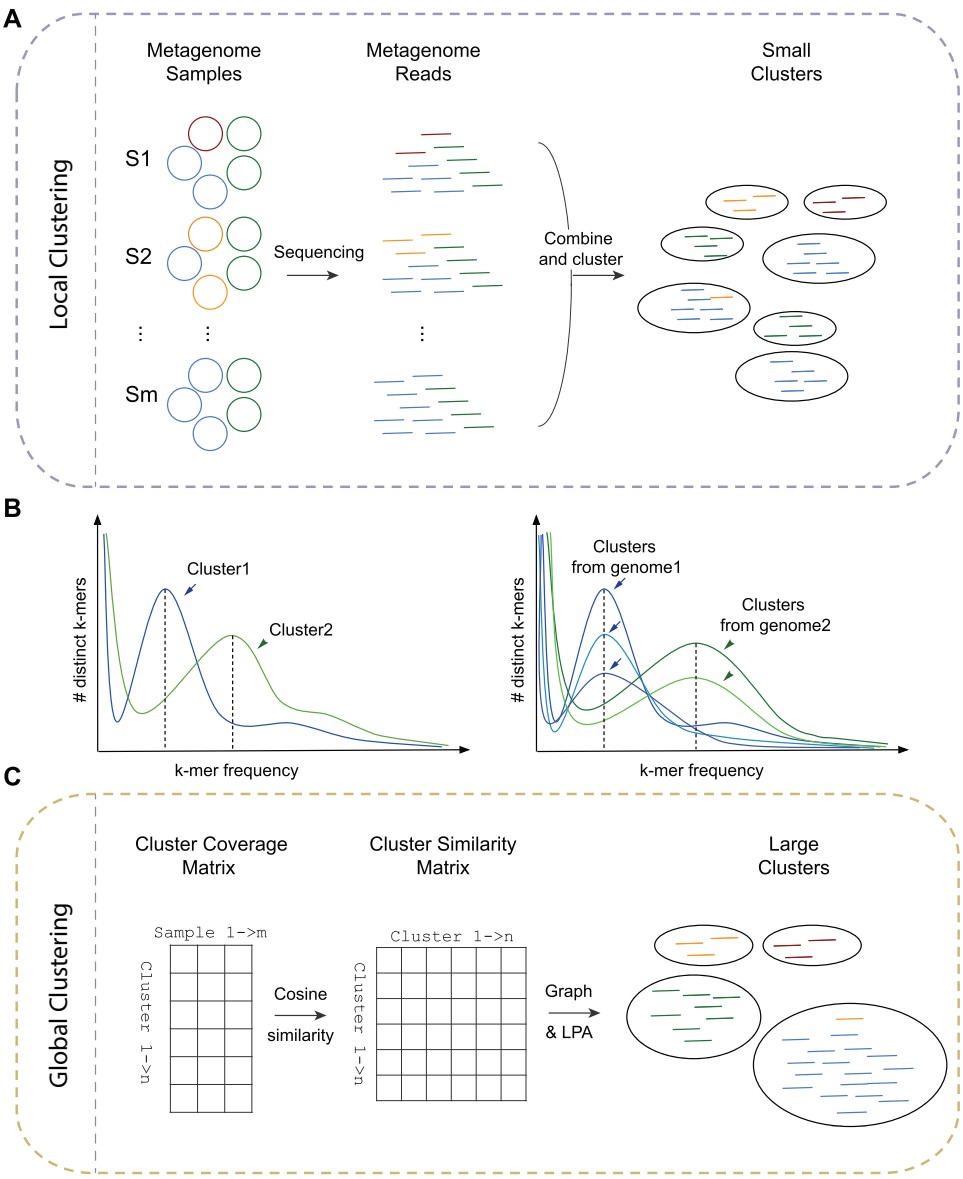

**Figure 1 An overview of the clustering strategies.** (A) Local clustering: short reads sequences from multiple samples of a microbial communities (such as derived from different sample sites or times, S1, S2…Sm) are combined and clustered using the scalable overlap-based clustering algorithm in SpaRC. Many small clusters are formed and reads from the same genomes scatter across many clusters (under-clustering). (B) Estimating genome coverage from unassembled read clusters. In the left illustration, two read clusters show different $k$-mer frequency peaks, each corresponding to the coverage of their underlying genome (dotted lines). In the right illustration, multiple read clusters derived from the same genome in theory will have the same genome coverage in a given sample, while the height of the peak (number of $k$-mers) can be very different depending on the size of the read clusters. (C) Global clustering. First, sequencing coverage of each small cluster from the local clustering step is estimated and a cluster coverage matrix is derived. Second, a square similarity matrix is obtained by computing pair-wise cosine similarities between all clusters. Finally, a graph is constructed using clusters as nodes and their similarity as weighted edges. Larger clusters containing all the reads from individual genomes can be obtained by partitioning the graph using the Label Propagation Algorithm (LPA).

in theory, will show different $k$-mer frequency peaks (genome coverage) in a given sample, while clusters from the same genome will show similar peaks, even though the number of $k$-mers at those peaks could be very different, with larger read clusters having more $k$-mers. We therefore sample several $k$-mers around the $k$-mer frequency peak of a cluster, and use the median of their counts among each sample to estimate the coverage of this cluster in each sample. We term these $k$-mers as representative $k$-mers for each cluster. Because very small clusters do not generate reliable $k$-mer spectra, we only select clusters with more than 50 reads for further clustering. We modified the original KMR function in SpaRC to track $k$-mer counts of each sample.

The cluster coverage information is stored in a $m$ by $n$ coverage matrix, where $m$ is the number of samples and $n$ is the number of read clusters.

### Calculating similarity between clusters

In the above cluster coverage matrix, every cluster is represented by a vector of counts. If two clusters are derived from the same genome, we expect that their vectors should be very similar. We chose cosine similarity to measure the similarity of cluster vectors as it is most commonly used in high-dimensional positive spaces. After all pairwise similarities are calculated, the coverage matrix is transformed into a $n$ by $n$ similarity matrix, where $n$ is the number of clusters. We only keep the similarity exceeding a predefined threshold because of the sparse nature of this matrix. This threshold parameter has a direct impact on clustering performance, as higher thresholds produce smaller but purer clusters. An optimal threshold parameter could be determined by performing a grid search for the one that gives the best clustering accuracy on a labeled dataset. For real-world metagenome datasets without known reference, this parameter has to be guessed.

### Graph construction and partitioning

By using the cosine similarity calculated above as weighted edges and the clusters as nodes, a weighted graph can be constructed. This cluster graph can then be partitioned into big clusters the same way as in the local clustering step by using the Label Propagation Algorithm (*Raghavan, Albert & Kumara, 2007*).

### Cluster assembly and quality evaluation

We selected large clusters (more than 1,000 reads for CAMI2 and more than 8,000 reads for MetaHIT) to assemble. Under these criteria, more than 90% of original reads were retained in these two datasets. These largeread clusters were assembled with MEGAHIT (ver 1.2.5-beta), (*Li et al., 2015*) using default parameters. The resulting contigs were binned with MetaBAT 2.0 (*Kang et al., 2019*) using default parameters. Smaller clusters were omitted from further analyses.

MetaQuast (ver 5.0.2) (*Mikheenko, Saveliev & Gurevich, 2015*) was used for metagenome assembly evaluation. As MetaHIT dataset does not have known references, we built a reference database by BLASTing the assembled contigs against NCBI nonredundant reference genomes, and selected the subset of references that have sequencing coverage of 30× or more ($n$ = 68). There were many bins from the MetaHIT

**Table 1 Datasets used in this study.**

| Dataset | CAMI2 | MBARC-26 | MetaHIT |
|---|---|---|---|
| # Samples | 64 | 1 | 228 |
| # Genomes | 791 | 26 | 161[a] |
| Read length (bp) | 2 × 150 | (90–150) × 2 | 75 × 2 |
| Total size (Gb) | 320 | 3.3 | 522 |

**Note:**
[a] Number of reference genomes the reads mapped to.

dataset having no genomes mapped to them, and they were omitted from further analyses. There were also bins mapped to multiple genomes, and genomes split into multiple bins. If the predominant part of a genome is included in a bin, then the completeness and purity of this genome were calculated according to all the contigs within this bin. Genome assembly quality evaluation metrics were obtained with MetaQuast using default parameters.

## Datasets

### The MBARC-26 microbial community

This mock dataset is a synthetic community with real-world sequence data (*Singer et al., 2016*). It contains Illumina reads from 23 bacteria and 3 Archaea species with known reference genomes. The sequence length is (90–150) × 2 bp totaling 3.3 Gb (Table 1). This dataset was used as a toy dataset for testing local clustering with minimizers.

### CAMI2 mouse gut metagenome dataset

The benchmark experiments on global clustering were done on a simulated dataset from the second CAMI Challenge (https://openstack.cebitec.uni-bielefeld.de:8080/swift/v1/CAMISIM_MOUSEGUT/). This dataset was simulated using 791 reference genomes derived from mouse gut microbiome, and it contains 64 samples with various genome coverage. Some relevant statistics of this dataset is shown in Table 1. A complete list of the organisms in this dataset is available at this URL (https://openstack.cebitec.uni-bielefeld.de:8080/swift/v1/CAMI_DATABASES/taxdump_cami2_toy.tar.gz).

### MetaHIT human gut metagenome dataset

To benchmark SpaRC on real-world datasets, we compiled a human microbiome metagenomic dataset from the MetaHIT project (https://trace.ncbi.nlm.nih.gov/Traces/sra/sra.cgi?study=ERP000108) by selecting 228 samples with a read length of 75 × 2 bp from the total 264 samples. These reads were mapped to a reference database and 161 reference genomes with at least 5× coverage were selected for assembly accuracy evaluation.

## Computing environments

Read clustering experiments were performed on Amazon Web Service (AWS)'s Elastic MapReduce (EMR, emr-5.17.0). Depending on the size of the dataset, a number of

**Table 2 Configuration of AWS EMR.**

| Parameter | Setting |
| --- | --- |
| # of cores/node | 8 |
| Memory/node | 61 |
| Storage/node | 300 GB SSD |
| Ethernet | 10 Gbps |
| Spark version | 2.3.1 |
| Hadoop version | 2.8.4 |
| Cluster mode | YARN |
| # of executors/node | 2 |
| Driver memory | 40 GB |
| Driver cores | 5 |
| Memory/executor | 24 GB |
| Cores/executor | 3 |
| HDFS block size | 32 MB |

r4.2×large instances were used to form a cluster, and the configuration details are shown in Table 2.

# RESULTS

## Global clustering greatly improves short read clustering performance

In order to test whether multiple samples derived from the same microbial community could be leveraged to improve short read clustering performance, we designed a control dataset by taking 10% of the reads from 50 samples from the CAMI2 synthetic metagenome dataset (Materials and Methods). The smaller dataset allowed us to reduce computation cost and obtain results faster. We clustered the reads using two clustering methods: in the "local clustering" method, we combined all the reads from the 50 samples for clustering and selected clusters with 50 reads or larger; in the "global clustering" method, we further applied the global clustering module to these clusters to form big clusters (Material and Methods). This labeled synthetic dataset enabled us to systematically compare the clustering performance between these two methods for cluster size, purity, completeness. The results are shown in Fig. 2. We used the same purity and completeness metrics as in *Shi et al. (2018)*. Briefly, the purity of a cluster is defined as the percentage of reads from the predominant genome within the cluster, while the completeness of a cluster is defined as the percentage of all the reads from the predominant genome that are captured by the cluster. Because almost identical strains from the same species were engineered in the dataset, both of the two metrics, especially purity, likely underestimate species-level clustering performance. For example, if a species has two closely related strains with equal number of reads, clusters derived from this species will have a purity of 50%. We therefore also measured cluster purity at the species level. In this experiment, the parameters were $k = 41$, $m = 22$, min_shared_kmers=2, max_degree=25, representative $k$-mer count = 100, and cosine threshold=0.925.

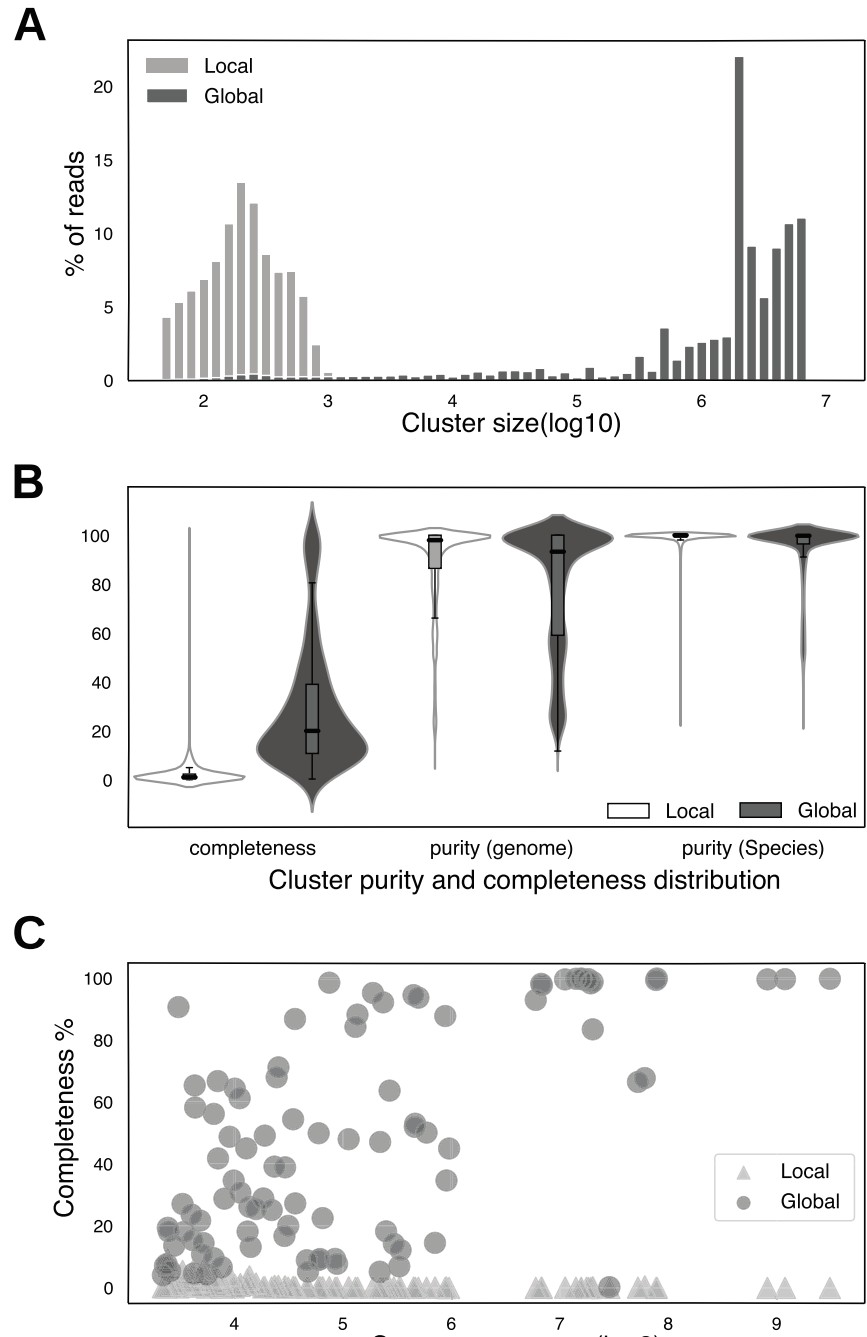

**Figure 2 A comparison of the clustering performance between local clustering and global clustering.**
(A) The distribution of clustered reads among the clusters for local clustering (light gray) and global clustering (dark gray). *X*-axis is cluster size (log10) and *y*-axis is the percent of reads that are clustered at a given cluster size. Cluster size refers to the number of reads in a cluster. (B) Violin plots of cluster completeness and purity (at genome level and species level). Global clustering metrics are plots filled in dark gray. The units on *y*-axis are percentages. (C) A scatter plot of sequencing coverage of the genomes and their completeness from local clustering (light gray triangles) and global clustering (dark gray circles). *X*-axis is the sequencing coverage (log2) and *y*-axis is the completeness in percentage.

The local clustering step resulted in 78.29% of the reads clustered into many small clusters ($n = 378,829$), with the largest cluster having only 10,151 reads. The majority of the clustered reads 99.08% are distributed in clusters with 1,000 reads or less (Fig. 2A). In contrast, after the global clustering step the number of clusters is significantly reduced ($n = 10,083$), 78.49% of reads are in clusters with 1,000,000 or more reads, and the largest cluster contains 6,836,687 reads. Consequently, the median completeness of the clusters from the global clustering is also 19.58 times larger than that from the local clustering (19.97% vs 1.02%, Fig. 2B). The increase in completeness came at the expense of some purity loss (median purity from 97.95% to 93.25% at the genome level%), but most of the clusters are still pure , especially at the species level (median purity decreased slightly, from 100% to 99.78%) (Fig. 2B).

As the success of genome clustering should heavily depend on its sequencing coverage, we next explored the relationship of cluster completeness as a function of genome sequencing coverage (Fig. 2C). For local clustering, higher sequencing coverage seems to have little effect on cluster completeness. In contrast, higher sequencing coverage does translate into higher completeness, suggesting global clustering effectively leverages multiple sample statistics for read clustering. After the sequencing coverage reaches a sufficient threshold (100×, Fig. 2C), the completeness of most genomes exceeds 80%.

## The performance of short read clustering can be improved by increasing the number of samples

We next explored the relationship between the number of samples in a dataset and the global clustering performance. Intuitively, more samples should enable a more robust estimation of the similarity between clusters and lead to better clustering performance. By limiting the total size of the datasets to 25 Gb to reduce the computation cost, we made several datasets with varying number of samples (5, 10, 20, 50) randomly selected from the CAMI2 synthetic metagenome dataset (Materials and Methods). We obtained clusters from these datasets by running SpaRC with the same parameters ($k = 41$, $m = 22$, min_shared_kmers=2, min_read_per_cluster=50) for local clustering, and representative_kmer_count=100 for global clustering.

As in the previous section, we evaluated the purity and completeness of the resulting clusters. As shown in Fig. 3, the median purity of the clusters at the genome level slightly dropped as the number of samples increases, from the highest 96.96% (at $n = 5$) to the lowest 88.89% (at $n = 20$, Fig. 3A). This is likely caused by the fact that more samples may contain more strain variation, and currently SpaRC can not distinguish very similar strains. Consistent with this notion, the purity measured at the species level remains largely unchanged (Table S1). In contrast, the completeness continuously increases with increasing number of samples (median completeness rises from 7.69% at $n = 5$ to 19.97% at $n = 50$, Fig. 3B). This result supports the hypothesis that more samples enhance global clustering performance, likely due to better estimation of cluster similarity.

The ultimate goal of read clustering is to recover the complete set of any genome without any contamination from other genomes. In order to measure how many genomes can be recovered by read clustering, here we define "a clustered genome" as a read cluster

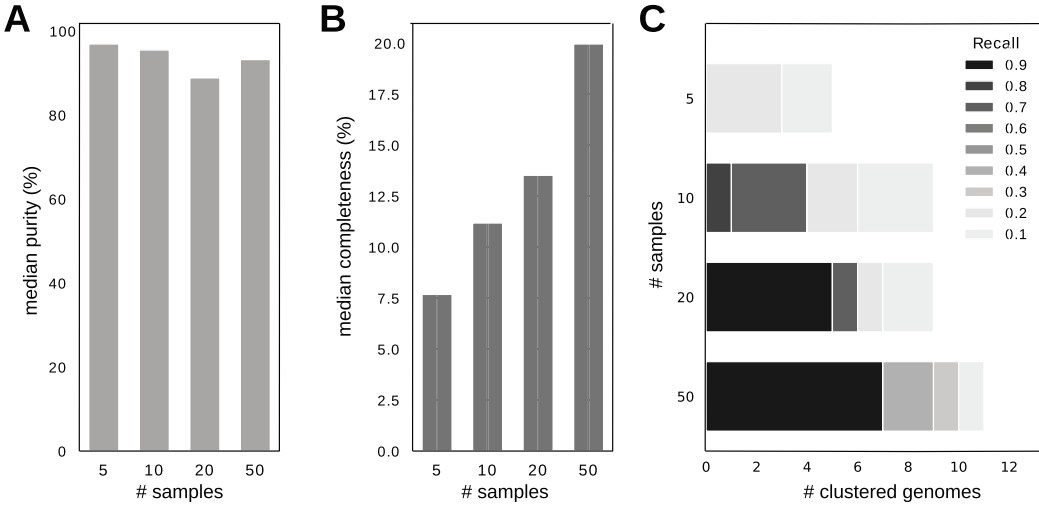

**Figure 3 Clustering performance with a different number of samples.** (A) Median purity comparison among a different number of samples. *X*-axis is the number of samples and *y*-axis is median purity. (B) Median completeness comparison among different number of samples. *X*-axis is the number of samples and *y*-axis is median completeness. (C) The number of clustered individual genomes with purity >95% and completeness >80% among different number of samples. There are 97 genomes with sequencing coverage >10× in this dataset. Different shades of gray represent different completeness levels. *X*-axis is the number of genomes and *y*-axis is the number of samples.

that simultaneously satisfies two criteria: purity >95% and completeness >80%. It is worth noting that these criteria are very strict. As strain-level heterogeneity can greatly reduce purity, and one small region larger than a read with no sequencing coverage, either due to statistical sampling or systematic sequencing biases, will greatly decrease completeness.

The clustered genomes from 5, 10, 20 and 50 samples are 0, 1, 5 and 7, respectively (Fig. 3C). Since the ability to obtain a clustered genome depends on the sequencing coverage, and there are 20 genomes with at least 100× coverage, this translates to a recovery rate of 35% with 50 samples. Other genomes with lower sequencing coverage also benefit from more samples included in clustering, as Fig. 3C shows the extent of recovery for all genomes with a sequence coverage >10×. The details of these 7 recovered genome can be found in Table S1.

## Parameters that may impact clustering performance

There are 10 parameters in the local clustering algorithm (*Shi et al., 2018*). In the global clustering step two more parameters are added, where the number of representative *k*-mers used to estimate cluster abundance (*rp*) and cosine similarity threshold (*cs*) to control graph complexity. While some of these parameters only affect computing efficiency, there are four parameters that in theory may affect clustering accuracy: *k*-mer(*k*)/minimizer(*m*) length, min_shared_kmers among reads (*minsk*), *rp* and *cs*. In theory higher *k*, *minsk*, and *cs* all lead to smaller clusters with lower completeness but higher purity, and *vice versa*. Higher *rp* should make the estimation of cluster abundance more accurate with a small cost in computing efficiency. To explore the effect of these

**Table 3  Clustering performance vs different parameters.**

| Parameter | | #Reads clustered | #Clusters | Median purity | Median completeness | #Clustered genome |
|---|---|---|---|---|---|---|
| *k*-mer length (*k*) | 31 | 58,400,733 | **9,324** | 82.10 | 16.15 | 2 |
| | 41 | **60,791,068** | 15,784 | 91.83 | 18.75 | **7** |
| | 51 | 58,333,152 | 14,265 | 92.61 | 23.00 | 7 |
| | 61 | 54,935,695 | 12,795 | **93.97** | **25.64** | 7 |
| Min_share_*k*-mers (*minsk*) | 1 | **60,953,377** | **13,401** | 91.26 | 20.57 | 7 |
| | 2 | 58,333,152 | 14,265 | 92.61 | 23.00 | 7 |
| | 3 | 54,991,601 | 14,836 | 94.40 | 26.30 | 7 |
| | 4 | 41,027,671 | 14,806 | 95.00 | 33.32 | 6 |
| | 5 | 26,643,712 | 13,598 | **96.33** | **42.78** | 7 |
| Representative *k*-mers (*rp*) | 9 | 60,791,068 | 15,784 | 91.80 | 18.75 | 7 |
| | 50 | **61,794,152** | 10,119 | 92.98 | **19.99** | 7 |
| | 100 | 61,736,159 | **10,083** | **93.25** | 19.97 | 7 |
| Cosine similarity threshold (*cs*) | 0.85 | **61,386,721** | **6,065** | 76.73 | **33.33** | 4 |
| | 0.875 | 60,689,758 | 10,347 | 88.24 | 31.04 | 5 |
| | 0.90 | 59,665,880 | 12,538 | **95.37** | 28.75 | 7 |
| | 0.925 | 58,408,305 | 13,570 | 93.13 | 25.90 | 7 |
| | 0.95 | 55,285,172 | 19,826 | 93.64 | 18.76 | 7 |

**Note:**
Numbers in bold indicate best results within each category.

parameters on clustering accuracy, we ran SpaRC with different sets of parameters on the 25 Gb dataset with 50 samples. In each of the parameter sets, only one parameter varies while the other three were held constant. The rest of the parameters were used as their default values. We used several metrics including number of reads clustered (bigger is better), number of clusters formed (bigger is worse), median cluster purity, median cluster completeness and number of clustered genomes to measure clustering performance. The results are shown in Table 3.

Overall, there is no single best parameter that can maximize all metrics. In general, the most abundant genomes (number of clustered genomes) are less likely to be affected by these parameters except a few extreme cases (where *k*-mers are set too small or cluster similarity thresholds are too low). In those extreme cases over-clustering happened, as only a small number of clusters formed with very low median purity. These parameters, except the number of representative *k*-mers, can greatly affect the median purity and completeness. Longer *k*-mers and requiring more shared *k*-mers among reads increases median purity and completeness. These improvements are likely driven by better clusters from the genomes with medium to high sequencing coverage. If these parameters became very large, the number of reads clustered dramatically decreases (under-clustering), but these genomes do not seem to be affected. For example, the number of clustered reads drops from 60,953,377 at *minsk*=1 to only 26,643,712 at *minsk*=5, but the median completeness and purity reaches their peak. The un-clustered reads at high *minsk* presumably are derived from a lot of genomes with low sequencing coverage, as

bigger *k*-mer size decreases the correct *k*-mers present in the dataset and makes their counts more noisy (*Chikhi & Medvedev, 2013*).

The number of representative *k*-mers used for cluster abundance estimation seems to have a very minor effect on clustering performance. If the number is too low, then the performance is slightly lower.

As expected, very low cluster similarity thresholds cause over-clustering, and very high ones lead to under-clustering. Same as the other parameters, the best parameter choice should be determined by the underlying scientific requirements to balance sensitivity and specificity.

## Assembly quality comparison between SpaRC-based and the classic approach

Compared to the current metagenome assembly strategy without clustering, that is, assemble the entire dataset followed by binning (hereafter referred as "the classical approach"), clustering the reads into individual clusters by SpaRC followed by assembly and binning (hereafter referred as "SpaRC-based approach") may produce better results. To test this hypothesis, we carried out both approaches on the above CAMI2 testing dataset (Methods) and compared their assembly results.

As shown above, the effectiveness of clustering depends on sequencing coverage, we therefore evaluated the assembly performance for genomes with coverage of 100× or above, 50–100× and 30–50×, respectively. We observed a comparable number of mis-assemblies from both approaches (Table S2), so we focused on the following four metrics: genome coverage (percent of reference covered by the assembled contigs), contamination (percent of contigs not belonging to the reference at the species level), N50 and L50 (measuring contiguity of the assembly). Specially, we only reported genome bins with at least 95% purity, or less than 5% contigs from other species. We evaluated their assembly accuracy in terms of near-complete genomes (95% genome coverage and above) and fairly complete genomes (80–95% genome coverage). The results are shown in Fig. 4.

At high sequencing coverage (100× or above), the SpaRC-based clustering approach was able to recover more near-complete genomes than the classical approach, 16 for SpaRC vs 11 for Classic (Fig. 4A). Among them, the SpaRC approach assembled 6 genomes that were not completely assembled by the classical approach, while missed only one assembled by the classical approach (Fig. 4B). The number of fairly complete genomes assembled by the two approaches are comparable at this sequencing coverage. Besides, 11 genomes assembled by the classical approach have smaller L50s and larger N50s while 6 genomes assembled by SpaRC approach do (Table S2). As sequencing coverage is getting lower, the classical approach has an advantage over the SpaRC-based approach, especially when coverage drops to below 50× (Fig. 4A). These genomes tend to spread across many small pure clusters. This result is consistent with the above clustering performance analyses, suggesting there is still an "under-clustering" problem for global clustering.

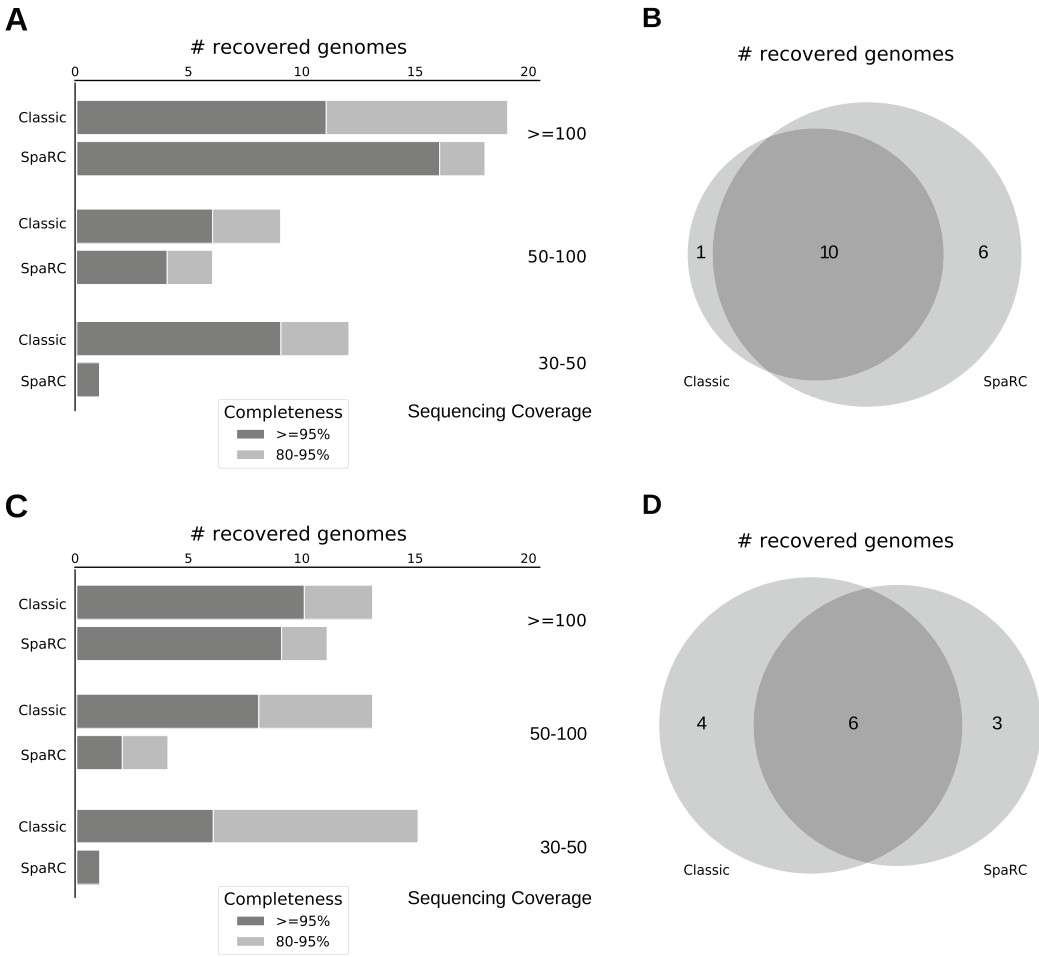

**Figure 4 Assembly accuracy comparison between two alternative strategies on CAMI2 simulated dataset (A and B) and the MetaHIT human microbiome dataset (C and D).** "Classic": The classical approach for metagenome assembly (MEGAHIT-MetaBAT), "SpaRC": Clustering-based assembly approach (SpaRC-MEGAHIT-MetaBAT). (A) Number of recovered genomes at 100× and above, 50–100×, 30–50× sequencing coverage, respectively. Recovered genomes are shown at two completeness levels. (B) Overlap between the near-complete genomes between the two approaches at sequencing coverage 100× and above. (C) Number of recovered genomes at 100× and above, 50–100×, 30–50× sequencing coverage, respectively. Recovered genomes are shown at two completeness levels. (D) Overlap between the near-complete genomes between the two approaches at sequencing coverage 100× and above.                  

We added a binning step using MetaBAT after the assembly of each of the clusters because some of the large ones are mixtures of a few genomes. Clusters with more than 100,000 reads contain 3.125 genomes on average, suggesting the species complexity of these clusters are much more reduced comparing to the original metagenome. The metrics presented in Fig. 4 were calculated after the binning step.

To test whether these conclusions can be generalized to real-world datasets, we applied the two approaches to a human microbiome dataset (MetaHIT, "Methods"). As there is no ground truth for this dataset, we mapped the metagenomic short reads to NCBI

non-redundant reference genome database, and used the 68 genomes with 30× or more sequencing coverage as a reference set.

Similar to the previous experiment on CAMI2 dataset, the SpaRC-based approach can cluster genomes with 100× sequencing coverage from MetaHIT dataset, and this ability decreases dramatically as coverage decreases. The SpaRC-based clustering approach was able to recover 9 near complete genomes with 100× or more coverage (total 16), while the classic method recovered 10 (Fig. 4C). Among them, SpaRC approach recovered 3 genomes that were not completely recovered by the classical approach, while missed 4 genomes (Fig. 4D). For a complete list of the recovered genomes from these two approaches on the two datasets, please refer to Table S3.

The above results suggest that adding a clustering step can complement the common metagenome assembly strategy, at least for the species with high sequencing coverage.

## Execution time of SpaRC on the MetaHIT dataset

Assembling large metagenome datasets using MEGAHIT requires a large amount of RAM on the server. It took 24.63 h on a single node with 64 CPU cores and 488 GB RAM (r4.16×large, a Memory-Optimized instance of AWS Elastic Compute Cloud) to assemble the MetaHIT dataset. As for SpaRC-based approach, we were able to distribute the read clustering step on an AWS Elastic MapReduce (EMR) cluster with 350 nodes, each with 8 CPU cores and 61 GB of RAM (Table 1). The clustering step took 8.9 h to complete. These results suggest that SpaRC is not as good as the classic approach in terms of costs and computational efficiency, instead it offers an advantage to scale up to bigger datasets (over assemblers only run on single nodes), and overall shorter computational execution time.

## DISCUSSION

Even with global clustering, there is still a lot of room left to further improve the accuracy of metagenome read clustering, as both under-clustering and over-clustering problems are still outstanding. One idea is to employ better metrics to improve the prediction that different clusters belong to the same species. The read clustering problem is similar to the metagenome binning problem. In the unsupervised metagenome binning problem, contigs are further clustered into genomes based on two metrics, tera-nucleotide-frequency (TNF) and abundance co-variation. TNF represents sequence composition biases among different species, and it is a useful metric to group contigs with similar sequence composition together during the binning process (*Kang et al., 2015*). Here we have already applied the abundance co-variation metric to improve clustering. The application of the TNF metric, however, is not straightforward, as TNF may not be reliably estimated from unassembled reads. Future analyses will be needed to integrate more information such as TNF into the clustering framework to reduce the requirement of many samples, given the fact that most of the metagenome shotgun sequencing experiments were carried out on single samples. The representative *k*-mer approach we used here is rather a naive one for efficient computing, but there are a few existing solutions that might be

leveraged to further improve the accuracy of cluster similarity calculation, such as the strategy in *Girotto, Pizzi & Comin (2016)*.

Another idea to improve the accuracy of metagenome clustering may be leveraging long read sequencing technologies. Long read technologies (such as PacBio and Nanopore) are increasingly applied to metagenome sequencing, a hybrid clustering approach leveraging long reads to cluster short reads may also greatly increase clustering performance. Long reads should be very helpful to increase clustering completeness as they span the regions where short reads have low coverage, and improve clustering purity at the strain level as they can distinguish repeats and closely related species or strains.

The global clustering step added more parameters to the entire clustering pipeline. We have shown that clustering accuracy can be influenced by some of these parameters, including $k$-mer/minimizer size, minimum shared $k$-mers to detect overlap, and abundance similarity thresholds among clusters to construct the cluster graph. Deriving an optimal set of parameters is challenging because it is likely dependent on the underlying data characteristics and the scoring metrics. Further, searching for an optimal parameter set from a large parameter space on a large dataset using grid-search or random-search strategies is computational prohibitive. There isn't an evident law to select the default parameters. This problem may be a good candidate for Bayesian optimization (*Snoek, Larochelle & Adams, 2012*; *Hutter, Hoos & Leyton-Brown, 2011*).

It was worth noting that while the global clustering procedure improves clustering completeness, but this comes at a small cost of lower clustering purity. This is a trade-off inherent to all clustering problems, and the above suggested potential improvements, including better metrics, longer reads and optimal parameters, may only improve completeness or purity, but not both. TNF can help completeness, while introducing impurity as TNF signals from smaller clusters tend to be noisy. Long reads can help link small clusters, but they are not useful to separate impure clusters because of their limited sequencing coverage. Larger $k$-mer/minimizer sizes, more $k$-mers/minimizers required for a valid overlap, larger abundance similarity thresholds can all lead to clusters with higher purity, but will inevitably also lead to lower clustering completeness.

Running SpaRC on very large metagenome datasets like the MetaHIT was still very challenging. In addition to requiring a large number of nodes, the memory overflow problem may occur during the execution when the number of executors per node or the number of cores per executor is not set properly. Some parameters may have a data-dependent nature and have to be manually experimented. Future work is needed for a data-driven approach for selecting appropriate parameters before carrying out large-scale experiments.

## CONCLUSIONS

In summary, we extended our previous work on the Apache Spark-based read clustering by exploiting species co-variation across different metagenome samples to improve clustering completeness. Using complex control datasets with many samples, we showed the global clustering algorithm can dramatically improve both cluster size and genome completeness with only short reads. Besides the benefit of scalability offered by the Spark

platform, the clustering-then-assembly strategy we presented here may also empower users to optimize the assembly process, such as trying different assemblers/parameters on individual species to achieve better genome coverage, strain resolution, etc. Even without these optimizations, we showed that this strategy can recover additional genomes missed from the classic approach.

## ACKNOWLEDGEMENTS

We thank Alex Copeland and Harrison Ho for critical reading of the manuscript.

### Funding

The work was supported by the National Natural Science Foundation of China (No. 61802246) and the 111 Project (No. D18003). Zhong Wang's work was supported by the U.S. Department of Energy, Office of Science, Office of Biological and Environmental Research under Contract No. DE-AC02-05CH11231. The funders had no role in study design, data collection and analysis, decision to publish, or preparation of the manuscript.

### Grant Disclosures

The following grant information was disclosed by the authors:
National Natural Science Foundation of China: 61802246.
111 Project: D18003.
U.S. Department of Energy, Office of Science, Office of Biological and Environmental Research: DE-AC02-05CH11231.

### Competing Interests

The authors declare that they have no competing interests.

### Author Contributions

- Kexue Li performed the experiments, analyzed the data, prepared figures and/or tables, authored or reviewed drafts of the paper, and approved the final draft.
- Yakang Lu performed the experiments, analyzed the data, prepared figures and/or tables, authored or reviewed drafts of the paper, and approved the final draft.
- Li Deng conceived and designed the experiments, authored or reviewed drafts of the paper, and approved the final draft.
- Lili Wang performed the experiments, analyzed the data, prepared figures and/or tables, and approved the final draft.
- Lizhen Shi performed the experiments, authored or reviewed drafts of the paper, and approved the final draft.
- Zhong Wang conceived and designed the experiments, prepared figures and/or tables, authored or reviewed drafts of the paper, and approved the final draft.

## Data Availability

SpaRC is under the BSD license and freely available at: https://bitbucket.org/berkeleylab/jgi-sparc/.

## Supplemental Information

Supplemental information for this article can be found online at http://dx.doi.org/10.7717/peerj.8966#supplemental-information.

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
