# Peer review of "Deconvolute individual genomes from metagenome sequences through short read clustering"

_PeerJ, doi:10.7717/peerj.8966_

## Round 0.1 · original submission · Major Revisions

Thank you for your submission. As you can see, the reviewers mostly see the merit of your work, but ask for more context and comparison to competing methods. I suggest you address the major comments of the first reviewer by more clearly distinguishing your work from assembly-based methods. Reviewers 2 and 3 ask for more discussion on what makes your approach unique compared to existing work, and in particular call for additional experiments on real data and comparison to often used methods such as MEGAHIT. These issues should be addressed well if you decide to submit a revision.

·

Basic reporting

the paper has a clear structure and is sufficiently self-contained. the language is mostly clear and unambiguous (there are just few typos: see below). figures and tables are clear and of sufficient quality.

unfortunately the authors seem unaware that the idea of using coverage in multiple samples to improve metagenomic binning appeared already in 2014, and is by now well understood: see e.g. [1] and the papers citing it, e.g. [2]. that line of research actually performs assembly (rather than binning) on the union of all samples, so it might even be more accurate in practice, although possibly slower. the approach in the submitted paper might thus still be competitive in terms of speed, space efficiency, or amount of parallelism allowed, but no experiment comparing it to [1,2] or similar approaches is described.

[1] Alneberg, Johannes, et al. "Binning metagenomic contigs by coverage and composition." Nature methods 11.11 (2014): 1144.

[2] Lu, Yang Young, et al. "COCACOLA: binning metagenomic contigs using sequence COmposition, read CoverAge, CO-alignment and paired-end read LinkAge." Bioinformatics 33.6 (2017): 791-798.


TYPOS
PAGE|LINE|OLD|NEW|COMMENT

1|10|”deferentially affect”|”differentially affect”|
2|2|”de bruijn”|”de Bruijn”|
2|8|”then assemble each cluster”| ”then assembles each cluster”|
2|17|”under-clustering”| |shouldn’t it be “over-clustering”?
2|11|”We refer the new”|”We refer to the new”|
2|-7|”neither space or time”|”neither space nor time”|
2|-7|”To improve the computing efficiency”| ”To improve computing efficiency”|
2|-3|”3.3-fold of speed-up”| ”3.3-fold speed-up”|
3|5|”under-clustering”| |shouldn’t it be “over-clustering”?
4|2|”we expect their vectors”| ”we expect that their vectors”|
4|6|”the sparsity nature”|”the sparse nature”|
4|11|”the same way in the local clustering”| ”the same way as in the local clustering”|
5|-1|”the number of sample increases”| ”the number of samples increases”|
7|3|”The detail of these 7 recovered genome can be found in the Supplemental Table 1”| ”The details of these 7 recovered genomes can be found in Supplemental Table 1”|
7|6|”number of representative k-mers are used”| ”number of representative k-mers used”|
7|13|”In each of the parameter set, only one parameter varies while the other three were held constant”| ”In each of the parameter sets, only one parameter varies while the other three are held constant”|
8|9|”improve both the cluster size”| ”improve both cluster size”|
8|21|”improve clustering purity strain”| ”improve clustering purity”|

Experimental design

the paper is concerned with the relevant problem of unsupervised read clustering in metagenomes: given a set of metagenomic samples with similar content, partition their reads into bins that are in one-to-one correspondence with the bacterial species in the samples. the paper merges all metagenomes into a single read set, it clusters such read set by iteratively merging two reads if they share enough k-mers, and finally it merges the resulting, pure, clusters, based on their frequency in different samples. specifically, for each cluster C and each sample S, the paper builds a k-mer histogram, and uses the second peak in the histogram as a proxy for the abundance of C in S. the abundance vectors of all clusters are then represented as a weighted graph, where the weight of an edge is the (thresholded) cosine similarity between the adjacent clusters. the graph is eventually partitioned using label propagation.

the research topic is within the scope of the journal. methods are described with sufficient detail.

Validity of the findings

the experiments are described thoroughly. conclusions are clearly stated and limited to the data. data and code are provided, so it seems to me that the experiments can be replicated.

Reviewer 2 ·

Basic reporting

Overall comments:
The paper is nicely written and has a clear structure. The proposed new step to the clustering method seems to make a great improvement. This naturally implies a trade-off between completeness and purity.

Results are assessed by considering purity and completeness at the genome level. However, as noted by the authors, strains from the same species may differ only slightly and are extremely difficult to distinguish. I’d love to see results where purity and completeness are determined at the species level rather than at the genome level. This would also make more sense: when clustering reads you’d often wish to cluster the reads that belong to the same species, rather than the same strain. Furthermore adding species-level results would show the validity of two statements in the paper (“Because almost identical … species-level clustering performance.” and “This is likely caused … very similar strains.”).

The discussion is very short. In my opinion it would largely benefit from a discussion of the method’s limitations, rather than only mentioning its benefits. For example, the authors do mention that the global clustering procedure improves clustering completeness, but do not mention that this comes at a cost of lower purity. This is an interesting trade-off (inherent to the clustering problem) that in my opinion should be discussed here.

The paper does contain quite a number of typos and issues with sentence structure, please revise or consider revising by a native speaker.

The figures and tables are very clear and easy to interpret.

Specific comments:
- On page 2 it is written that "clustering metagenome reads based on k-mer statistics across samples is very challenging". Has anybody shown this? Please substantiate with references. The same holds for "The application of … for unassembled reads" in the discussion.
- I do not understand the first sentence of the section “Estimating cluster coverage”. Please revise.
- Description of the MBARC-26 dataset: “real-world”. How can the data be real-world if the dataset is synthetic? Did you mean simulated?
- I cannot find results on the MBARC-26 dataset. Is the dataset actually used?
- “Short read clustering performance is greatly improved with multiple samples from the same microbial community.” This header does not match the contents of the paragraph. Also the first sentence of the paragraph does not match the results presented here.
- Page 5: “without a big drop”: a drop of 10% is big, so please do not say it is not. However this big drop is not a problem: there is simply a trade-off to be made between completeness and purity.
- Page 5: “In contrast, higher … for read clustering.” I do not understand how higher sequencing coverage translating to higher completeness suggests this. Please explain.
- Page 7: “The unclustered reads … presumably … low sequencing coverage.” Can you verify this?
- Page 8: “This problem may … for Bayesian optimization.” Why would Bayesian optimization be a good idea here, and not something else? Please motivate.

Minor comments:
- abstract line 2: "Clustering short reads" please add "by species" or “by genome” (dependent on what your focus is)
- abstract: deferentially -> differentially
- page 2: “data not shown”. Why not? Please consider including it in the supplementary material.
- Figure 2B: add units to the vertical axis.
- Figure 2 caption: “x-axis is cluster size” measured as number of reads, number of bps?

Experimental design

The experimental design is well carried out and results are presented in a nice way. However I do feel that the experiments are rather limited: only one dataset is used (from the results section I deduce that the MBARC-26 dataset was not used). I'd like to see more experiments on different datasets. Besides, it is unclear how this approach performs compared to other read clustering methods. In my opinion the paper would largely benefit from a comparison of methods.

Validity of the findings

The results are clearly presented. The CAMI mouse gut data is publicly available. There is a bit too much speculation and the results that are not so supportive of this method (a reduction in purity) are ignored in some parts of the paper (see comments in block 1).

Reviewer 3 ·

Basic reporting

No comment.

Experimental design

No comment.

Validity of the findings

No comment.

Additional comments

Li et al. here extend a method that uses k-mer abundance correlations
across metagenome samples to cluster out "read clouds" based on
co-abundance. These read clouds should represent bacterial/archaeal
genomes.

This tool is presented against the dominant "MAG" background of
approaches to genome binning, where metagenomes are first assembled
and then the resulting contigs are binned into "metagenome assembled
genomes" based on abundance, tetra-nt frequency, etc. Given the
widespread utility of the MAG approach in recent years, we think the
authors should spend more time comparing and contrasting the utility
of their approach vs the dominant MAG approach. This would put the
paper in a fuller context.

Thoughts/comments:

The authors present their tool's performance on synthetic data sets,
but not on any real data sets. This limits the power of the
presentation.

There are good reasons to go with read clustering over assembly, and
they could be presented more thoroughly. Assembly by e.g. MEGAHIT is
not that memory challenging in practice, so the authors' interest in
claiming decreased memory requirements is not convincing on its own.
Additional points could be mentioned, such as being able to assemble
the reads with multiple assemblers, rapidly trial different assembly
parameters for each cluster, and avoid strain confusion (where strain
variation interferes with assembly.

Conversely, there are reasons to maybe go with assembly first, such
as the (potential) ability to recover parts of genomes that have low
or high abundance (strain variation, repetitive elements). Some of these
are mentioned in the paper already.

The authors should do a comparison against megahit for the reads that
are left out. What is this method capturing that megahit misses? What
does megahit capture that this method misses? What do they both miss,
on data sets where there is ground truth?

Why do some genomes with very low coverage (e.g. 4) have very high
completeness while others do not? What is biologically/computationally
different about them (e.g. divergent k=4 abundance?)?

Detailed questions/requests --

Please archive the version of software used in this paper and provide
a DOI; bitbucket is not archival.

What is the license under which this software is released? Please
specify in repository, and also in paper.

What is the definition of a reliable k-mer? (page 2, paragraph 3)
High abundance?

"they are likely to have a uni-modal k-mer spectrum, where the main
peak after the minimum inflection point represents the sequencing
coverage of the genome region where these reads are derived." This will
eliminate some strain variants, correct? Is this considered a positive,
a negative, or a user choice?

The tool is hard to work with and hard to install, it seems. It would
be ideal to provide some kind of docker container or cloud tutorial.

"Furthermore, this approach can potentially recover almost the full
set of reads of microbial genomes with sufficient representation in
the data, an ability influenced by several parameters including the
number of available samples.": This statement is very vague; could it
be sharpened, please?

Typos --

'deferentially' should probably be 'differentially'

The word 'unseen' is used (2nd paragraph), maybe this should be 'are
not known'? Or have not been cultured?

It should be MEGAHIT, not MegaHIT.

50, 10, 20 an 50 samples => 'and'

---

## Round 0.2 · Minor Revisions

As you can see, the reviewers raise some relatively minor issues, that can probably be dealt with easily by textual changes. Please add a short response to the points of the second reviewer in your resubmission.

·

Basic reporting

CONTENT
PAGE|LINE|OLD|NEW|COMMENT

7|250|"To test this hypothesis"||i don't think you are testing the hypothesis that the assembler can "optimize the assembly process of each individual genome": this sounds like the assembler automatically tuning its parameters or algorithms to each cluster.
8|293|"r4.16xlarge"||what is this?


TYPOS
PAGE|LINE|OLD|NEW|COMMENT

4|116|"those have"|"that have"|
4|120|"maped"|"mapped"|
4|120|"contains dominant part"|"contains the dominant part"|
8|274|"On average clusters"|"Clusters"|
8|278|"we applied the two approaches on"|"we applied the two approaches to"|
8|308|"there are a few existing solutions might be leveraged"|"there are a few existing solutions that might be leveraged"|
13|Panel B|"median ompleteness"|"median completeness"|
14|1|"alliterative"||?!

Experimental design

ok

Validity of the findings

ok

Reviewer 2 ·

Basic reporting

See report.

Experimental design

See report.

Validity of the findings

See report.

Annotated reviews are not available for download in order to protect the identity of reviewers who chose to remain anonymous.

---

## Round 0.3 · accepted · Accept

Thank you for carefully addressing the last remaining reviewer issues.